# G-quadruplex stabilization provokes DNA breaks in human *PKD1*, revealing a second hit mechanism for ADPKD

Agata M. Parsons, Seth Byrne ⓘ, Jesse Kooistra ⓘ, John Dewey ⓘ,
Aaron L. Zebolsky, Gloria Alvarado ⓘ, Gerrit J. Bouma,
Gregory B. Vanden Heuvel & Erik D. Larson ⓘ ✉

The "secondhit" pathway is responsible for biallelic inactivation of many tumor suppressors, where a pathogenic germline allele is joined by somatic mutation of the remaining functional allele. The mechanisms are unresolved, but the human *PKD1* tumor suppressor is a good experimental model for identifying the molecular determinants. Inactivation of *PKD1* results in autosomal dominant polycystic kidney disease, a very common disorder characterized by the accumulation of fluid-filled cysts and end-stage renal disease. Since human *PKD1* follows second hit and mouse *Pkd1* heterozygotes do not, we reasoned that there is likely a molecular difference that explains the elevated mutagenesis of the human gene. Here we demonstrate that guanine quadruplex DNA structures are abundant throughout human, but not mouse, *PKD1* where they activate the DNA damage response. Our results suggest that guanine quadruplex DNAs provoke DNA breaks in *PKD1*, providing a potential mechanism for cystogenesis in autosomal dominant polycystic kidney disease specifically and for the inactivation of guanine quadruplex-rich tumor suppressors generally.

The tumor-blocking phenotypes for most tumor suppressor genes are only revealed once a germline pathogenic allele is accompanied by inactivation of the remaining functional allele in a somatic cell, a pathway known as "secondhit"[1]. It is unknown why many tumor suppressors follow this pathway, or why orthologous genes in animal models are often genetically stable, but the human *PKD1* (h*PKD1*) gene presents a unique experimental opportunity to uncover a mechanism. *PKD1* encodes polycystin-1, a multi-functional transmembrane protein involved in cell proliferation, differentiation, apoptosis, cell adhesion, and fluid secretion. In ADPKD the single functional *PKD1* allele undergoes somatic second hit inactivation[2–7], initiating cysts and cyst progression due to loss of polycystin-1 activity[8–10]. Although mice have served as valuable animal models, mouse *Pkd1* heterozygotes do not faithfully recapitulate human cystogenesis and second hit[2–7]. This implies that an intrinsic

molecular feature influences h*PKD1*'s risk of DNA breaks, mutagenesis, and subsequent inactivation.

Several observations suggest h*PKD1* is mutation-prone compared to mouse *Pkd1* (m*Pkd1*). First, of the hundreds of renal cysts containing inactivated h*PKD1* alleles in an affected kidney, each somatic inactivation is derived from an independent h*PKD1* loss of heterozygosity or mutation event[3,4,6,7,11]. Furthermore, h*PKD1* is itself polymorphic and was partially duplicated in the human genome to produce 6 nearby pseudogenes[3,4,6]. One clue to the sources of h*PKD1* instability was the identification of a ~2 kb intronic sequence repeat in intron 21 that inhibits replication[12–16], sequences that are notably absent in m*Pkd1*[17]. This raises the possibility that DNA damage in h*PKD1* arises because of non-canonical DNA structure formation[12–16]. Consistent with that hypothesis, an 88 nucleotide repeat from intron 21 (IVS21) was reported to form guanine-quadruplex (G4) DNA[16,18], a four stranded

Department of Biomedical Sciences, Western Michigan University Homer Stryker MD School of Medicine, Kalamazoo, MI, USA.
✉e-mail: erik.larson@wmed.edu

DNA conformation (Fig. 1a). G4 DNAs are physiological structures found concentrated at regulatory domains in the genome, but if left unresolved they promote oncogenesis by inhibiting replication or repair to create DNA breaks that then lead to recombination and mutations[19–21].

## Results

### G4 sequences in *PKD1* orthologs

Based on the potential for h*PKD1* to adopt G4 DNA in intron 21 (IVS21)[16], we quantitated G4 DNA sequences in the entire *PKD1* gene using a G4-prediction program (QGRS mapper[22]). Using a conservative G4 motif definition of at least three tandem guanines repeated four times (Fig. 1a), we find high G4 DNA content in h*PKD1* (124) and low G4 DNA abundance in m*Pkd1* (13) and rat r*Pkd1* (10) (Fig. 1b). The predicted G4 DNA-bias in h*PKD1* compared to m*Pkd1* (Fig. 1b) agrees with experimentally validated cellular G4 DNAs (from G4-ChIPseqs) curated by the EndoQuad database[23]. Mapping of predicted G4 DNA onto h*PKD1* shows a broad distribution, with some introns showing clusters of G4 DNAs, including 16 in IVS1, 38 in IVS21, 7 in IVS22, and 5 in IVS42 (Fig. 1c). This contrasts with m*Pkd1* and r*Pkd1*, which have very few G4 motifs overall (Fig. 1b, c). The intronic G4 clusters found in h*PKD1* are composed of poly-purine repeats. A G4 motif from the 16 tandem repeats in intron 1 formed G4 in vitro as detected by dot blot (Fig. 2a) and shows a characteristic G4 spectrum by circular dichroism spectroscopy with a peak at 264 and a dip at 240[24,25], which shifted when the guanine repeats were disrupted by substitution (Fig. 2b). CD scans of similar guanine-rich repeats derived from IVS21, 22 and 42 also show G4 spectra (Supplementary Fig. 1), as expected. We conclude that G4-forming sequences are abundant and distributed throughout h*PKD1* but are comparatively rare in m*Pkd1* and r*Pkd1*.

### G4 DNA formation at the h*PKD1* locus

If G4 DNAs are relevant to ADPKD pathophysiology we would expect to find the structures in renal cell nuclei and at the h*PKD1* locus. The BG4 antibody has been well characterized for the visualization of quadruplexes in cells[26], and the newer SG4 camelid nanobody has low nanomolar affinity for G4s with a control version (R105A) available that

is mutated for G4 DNA specificity[27]. In our hands, all three antibodies performed as expected for G4 detection (Supplementary Fig. 2). To visualize the h*PKD1* locus in cells we employed CASFISH[28], since the denaturing conditions for Fluorescence In Situ Hybridization would likely disrupt DNA structures. In CASFISH, a catalytically inactivated CAS9, dCAS9, is fluorescently labeled and targeted to a locus with sgRNAs. Using a labeled dCAS9 paired with sgRNAs directed to the 3' end of h*PKD1* and combined with SG4 labeling, we found overlapping foci in normal (Fig. 3A, B), and ADPKD tissue (Fig. 3C, D). Substituting SG4 for the mutant nanobody (R105A) eliminated G4 signals (Fig. 3E, F) and omitting the sgRNAs ablated *PKD1* foci (Fig. 3G, H), confirming that the quadruplex signals are due to SG4 and that the *PKD1* signals are dependent on the sgRNAs. Thus, the G4 sequences in the h*PKD1* gene may adopt G4 structures.

We next used a more quantitative approach to compare G4 DNA formation in human and mouse *Pkd1* by employing chromatin immunoprecipitations (ChIPs) with the BG4 antibody, anticipating enrichment of human but not mouse *Pkd1*. Fragmented crosslinked chromatin from human embryonic kidney (293T) or mouse mIMDC3 was precipitated with BG4 followed by qPCR to detect template enrichment. For HEK293T, we compared a region adjacent to IVS21 to another intron, IVS34. Exon/Intron 34 has 2 potential G4 motifs compared to the 38 in intron 21 (Fig. 1c), making this region a good negative control. For mIMCD3 chromatin, we used primers specific to nucleotide positions comparable to h*PKD1*, which in m*Pkd1* are near IVS21 and IVS37. BG4-IP enriched for the G4-dense human IVS21 locus by a factor of 1.7 compared to IVS34 (Fig. 4). Addition of the G4 DNA-stabilizing ligand Phen-DC3 to HEK293T significantly ($P < 0.001$) increased IVS21 enrichment compared to vehicle by 3 times (16.6/5.6), and to IVS34 by 4.4 times (16.6/3.8) (Fig. 4). Repeating the experiment with a chemically unrelated G4-ligand, CX-5461, yielded similar results (Fig. 4). None of the m*Pkd1* loci were significantly enriched for either ligand ($P > 0.05$)(Fig. 4). Since a G4-rich control locus in mIMCD3 was enriched when either ligand was present (Supplementary Fig. 3), both ligands function in mIMCD3 and BG4-IP results are consistent with the lack of G4 DNAs in m*Pkd1* (Fig. 1b, c). Collectively, the data suggest that G4 DNA forms within h*PKD1*.

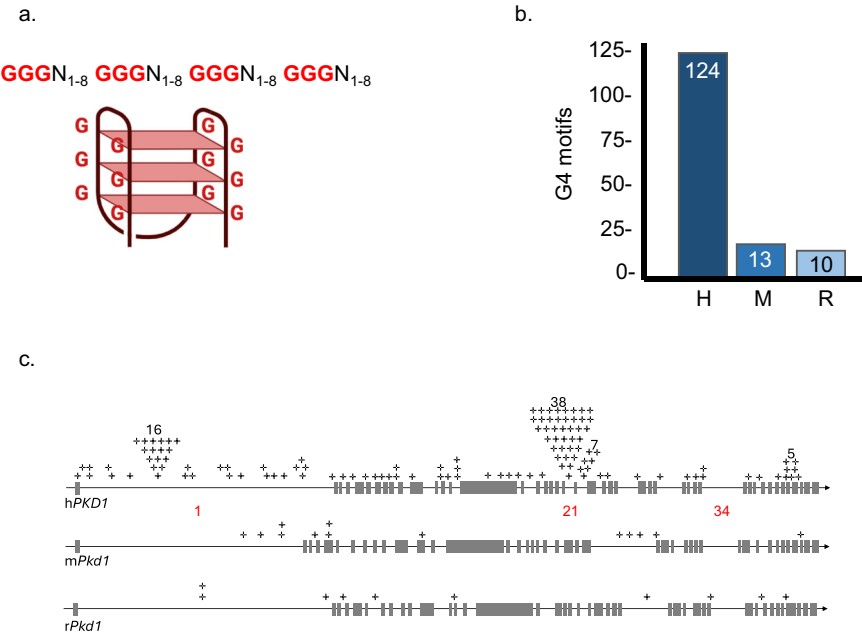

**Fig. 1 | G4 DNA sequences in *PKD1*. a** A basic G4 DNA sequence motif and model of a G4 structure. Guanines (G) pair with one another to form stacks of tetrads. Created in BioRender. Bouma, J. (2024) BioRender.com/n41z770. **b** G4 motifs quantitated for human (H), mouse (M) and rat (R) *PKD1* with QGRS mapper[22], using a 45 nt. window, 3G minimum, and 8 nt. loop. **c** G4 motifs (+) mapped onto the sense strand of human, mouse, and rat *PKD1*. h*PKD1* intron 1, 21, and 34 are marked (red text).

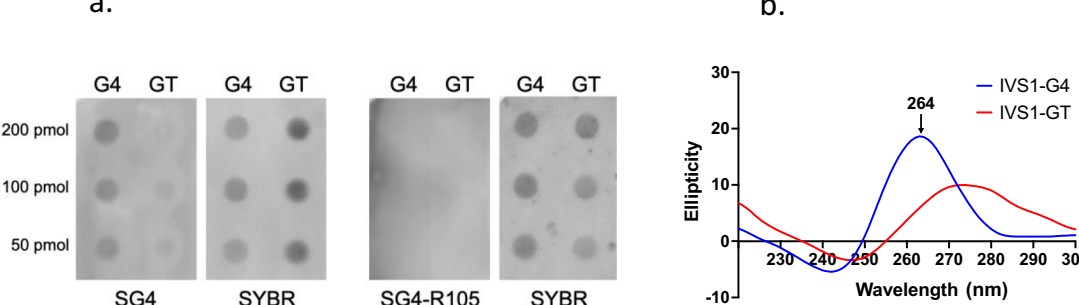

Fig. 2 | **G4 formation in vitro. a** Dot blot assay using SG4 and mutated SG4 (SG4-R105A) nanobodies on a G4-folded oligonucleotide from h*PKD1* intron 1 (G4) or thymine substituted (GT) control. Membranes were post-stained with SYBR gold (SYBR). **b** Circular dichroism (CD) spectroscopy of a representative G4 DNA repeat from IVS1 (G4) and a control DNA substituted to disrupt G4-folding potential (GT). A peak near 260 nm and dip at 240 nm indicates G4 DNA.

## Influence of G4 DNA on *PKD1* expression

The abundant G4 DNAs distributed throughout h*PKD1* (Fig. 1c) implies a functional role for the structures. Therefore, we asked if enforced G4 stabilization with a G4-specific ligand would alter *PKD1* expression. Both HEK293T and mouse mIMCD3 cells were treated with 10 μM Phen-DC3, cDNA was collected over time, and resulting expression quantified by qPCR. Results were normalized to expression of a non-G4 control, β-actin. h*PKD1* mRNA abundance was significantly reduced by more than half compared to m*Pkd1* after 2 days and by 3.25 times after a week (Fig. 5) ($P < 0.01$). Polycystin-1 protein was also reduced over this time as judged by Western (Supplementary Fig. 4). We cannot exclude the possibility that h*PKD1* transcription factors could be downregulated by G4 stabilization, but identification of those factors would be needed a priori. Even so, the abundance of G4 sequences in h*PKD1* (Fig. 1b) is consistent with G4-specific regulation in cis. While Phen-DC3 reduced polycystin-1 mRNA (Fig. 5) it should be noted that naturally folded G4s may impact various pathways, such as alternative splicing, so further studies on G4-based regulation of h*PKD1* are required to uncover the mechanisms.

## G4-induced DNA breaks in h*PKD1*

We reasoned that if G4 DNAs increase the risk of *PKD1* inactivation then G4 DNA stabilization should provoke genotoxic lesions. Upon DNA break formation, histone H2AX becomes phosphorylated at serine 139 (called γH2AX), generating up to a megabase-sized marker for signaling the DNA damage response[29]. Therefore, we used an anti-γH2AX antibody to assess G4 DNA-induced breaks by ChIP after Phen-DC3 treatment. Since γH2AX signals can be large[29], we used a locus on another chromosome, *PCNA*, as a non-G4 control[23]. γH2AX ChIP enriched for h*PKD1* (IVS21) compared to h*PCNA* (1.9/1), which increased to 3.7 times (5.2/1.4) ($P < 0.001$) when Phen-DC3 was added to cells (Fig. 6a). Neither m*Pkd1* nor m*Pcna* was enriched by γH2AX ChIP (Fig. 6a), consistent with a lack of G4 DNA at those loci. Experimental repeats with CX-5461 yielded similar results (Fig. 6a), indicating that the DNA breaks are a product of G4 formation and not due to the ligand itself. It is feasible that the γH2AX signals originated at regional G4s located adjacent to h*PKD1*, since h*PKD1* resides in a CpG-rich area of human chromosome 16[30], but ChIPs for RAD51, an essential recombination repair protein, resulted in a similar G4-dependent enrichment of h*PKD1* with no enrichment of h*PCNA* observed (Fig. 6b). Thus, G4 DNA folding causes DNA breaks in h*PKD1* and recombination repair activities respond to those lesions. We conclude that the differences in h*PKD1* and m*Pkd1* stability may be explained by the presence of G4 DNAs in the former and absence of the structure in the latter. Since DNA breaks are precursor lesions for known *PKD1* gene inactivation mechanisms, such as loss of heterozygosity[5–7] and gene conversion[31], formation of G4 DNAs in h*PKD1* provide a molecular rationale for second hit mutagenesis in ADPKD (Fig. 7).

## Discussion

G4 DNA is a biologically active structure that is well-known to regulate programmed recombination and gene expression[19–21,32], so it is likely that the poly-purine repeats in h*PKD1* have regulatory roles. Reduction of polycystin-1 below a certain threshold results in cysts[33], making it important to understand how G4 DNA formation influences polycystin-1 levels and its isoforms. The various mechanisms proposed for G4 DNA-modulated transcription include the recruitment of chromatin remodelers, alteration of methylation status, juxtaposition of distant elements via loop formation, or nucleation of liquid-liquid phase separation[20], and so G4 DNA motifs throughout h*PKD1* may have been retained as structural features for modulating more than one potential regulatory pathway. The effects may also be positive in nature, as G4 DNA folding in the c-*MYC* promoter has been shown to upregulate expression[34]. Thus, we propose that the G/C-rich sequences in h*PKD1* impart regulatory benefits to the gene that hinge on G4 DNA, R-loop[35] and/or H-DNA[14,15] structures, rather than the sequences themselves, but that it comes at some cost to gene stability[19–21,32].

Based on the model (Fig. 7), one might also predict increased renal cystogenesis in patients with deficiencies in G4-specific helicases, like BLM or FANCJ, however loss of those G4 DNA resolution activities results in severe phenotypes that may preclude identification of clinically significant renal cysts. Yet, given the essential nature of polycystin-1 and its diverse functions, deficiencies in G4 resolution activities could have developmental consequences via *PKD1* deregulation that merits investigation. Regardless, identification of G4 DNA in human but not mouse *PKD1* provides a mechanism for second hit gene inactivation that helps explain the autosomal dominant inheritance pattern for ADPKD.

The model that h*PKD1* second hit mutagenesis derives from G4 DNA-induced DNA breaks (Fig. 7) enjoys support from prior research on intron 21 showing that the poly-purine repeats from that region promote mutagenesis[4,14,15]. In addition, replication assays using an 88 nt. ectopic repeat from IVS21 caused replication-dependent DNA breaks and genetic instability[14,16,18,36], with the G4-binder telomestatin increasing deletions and hypermutation[16,18]. Considering that the DNA damage response is activated at h*PKD1* upon Phen-DC3 or CX-5461 exposure (Fig. 6), stabilization of G4 DNAs is likely responsible for causing the double-strand breaks. Since the responding repair pathways may initiate strand excision for repair, mutations that arise as a result can be distant from the G4-induced break site, making it difficult to identify the responsible G4 motif. Plus, G4 sequences are numerous and widely distributed in h*PKD1* (Fig. 1c). Still, that observation may explain the absence of mutation hot spots[7,11,37], and individual somatic second hit mutations identified in cyst sequencing studies[7,11] are just as widely distributed in h*PKD1* as the G4 DNA motifs (Supplementary Fig. 5). Despite that, and interestingly, there are similar independent somatic mutations that flank G4 DNAs in h*PKD1*, particularly

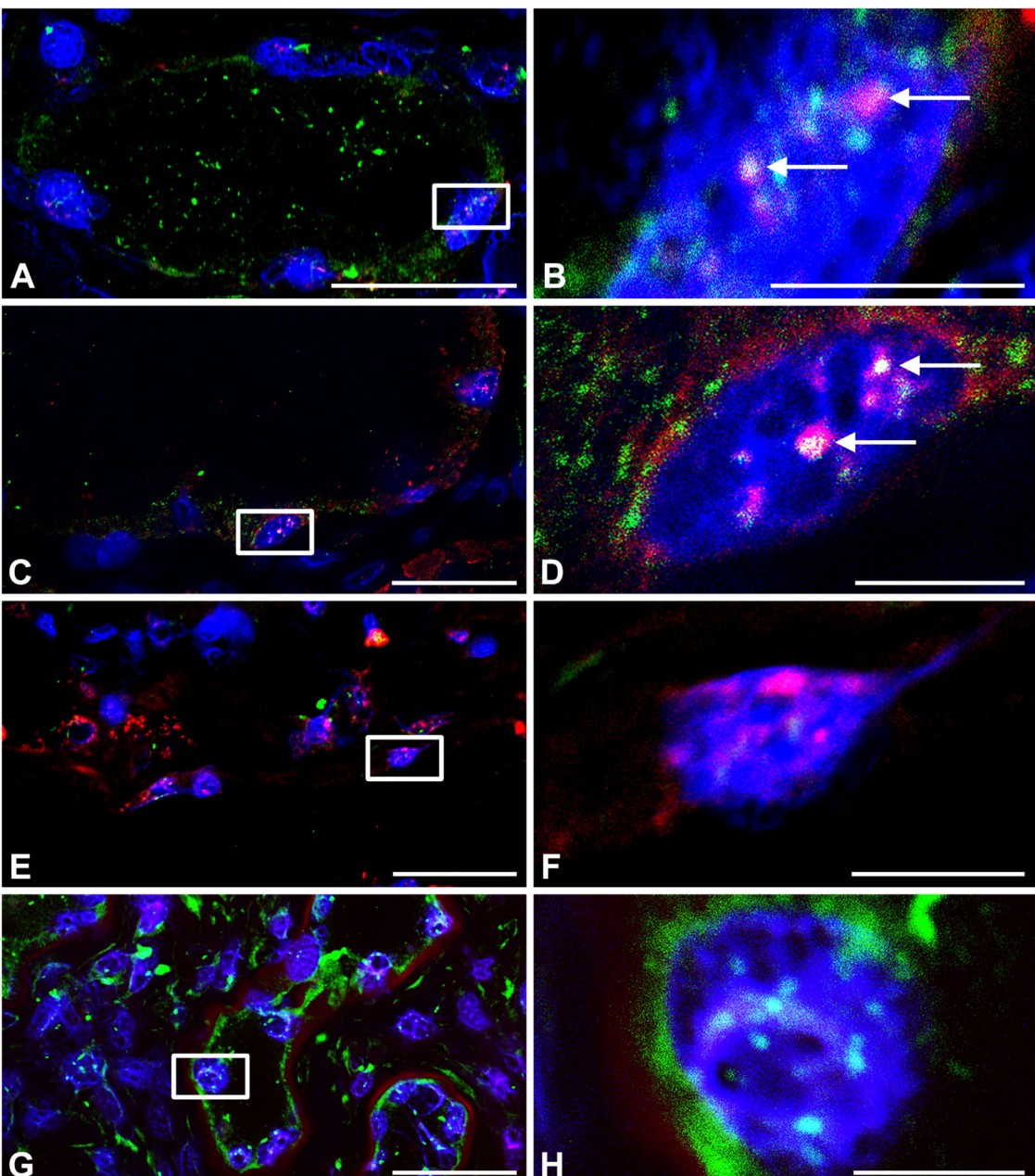

**Fig. 3 | G4 DNA at the human *PKD1* locus in normal and ADPKD tissue.**
**A, B** Normal human kidney tissue sections labeled with SG4 nanobody (green) and dCAS9 with sgRNAs to *PKD1* (red). **B** Boxed region in (**A**) enlarged showing G4 DNA and *PKD1* colocalization (arrows). **C, D** Human ADPKD tissue sections were labeled with SG4 nanobody against G4 DNAs (green) and dCAS9 with sgRNAs to *PKD1* (red). **D** Boxed region in (**C**) is enlarged to show G4 DNA and *PKD1* co-localization (arrows). **E–H** are controls. **E, F** normal human kidney tissue sections labeled with mutated nanobody (SG4mut-R105A) (green) and dCAS9 with sgRNAs to *PKD1* (red). **G, H** human ADPKD tissue sections labeled with SG4 nanobody (green) and dCAS9 without sgRNAs (red). Boxed region in (**E, G**) are expanded in (**F, H**), respectively. Scale bars in (**A, C, E, G** = 50 microns). Scale bars in (**B, D, F, H** = 10 microns).

surrounding intron 42[7,11] (Supplementary Fig. 5). Conclusions on the casual relationships and which G4 DNAs are the most mutagenic awaits development of a tractable model for measuring G4-induced second hits.

Stabilization of G4 DNAs has been a strategy for cancer drug development, with CX-5461 showing efficacy in patients with homologous-recombination deficient tumors[38,39] and increased mutagenesis in cultured cells[40]. Based on the potential for G4 DNA to provoke *PKD1* inactivation events, an approach that instead *destabilizes* G4 DNA could feasibly prevent or delay ADPKD onset. There are indeed small molecules identified that can disrupt G4 folding[41,42]. The treatment implication for h*PKD1* is that destabilization of G4

DNAs within *PKD1* would decrease the risk of inactivating mutagenesis and thus limit cystogenesis for at-risk individuals. This approach may have utility beyond ADPKD if G4 DNAs also influence the stability of other tumor suppressors. For instance, mice heterozygous for pathogenic *Brca1* mutations are not prone to spontaneous tumors, contrasting with humans who show *BRCA1* second hits and cancer predisposition. Correspondingly, G4 DNA appears to be more abundant in human *BRCA1* compared to mouse *Brca1*[23]. Therefore, it is possible that disease risk for individuals inheriting a pathogenic and G4-rich tumor suppressor allele could be mitigated through G4 destabilization strategies that lower the likelihood of second hit mutagenesis.

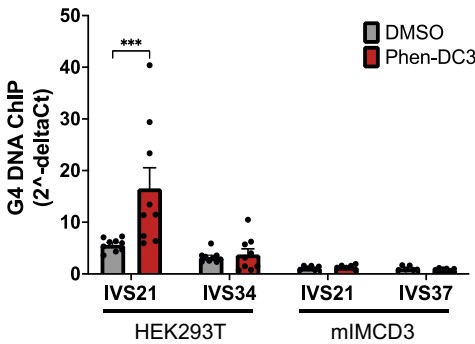
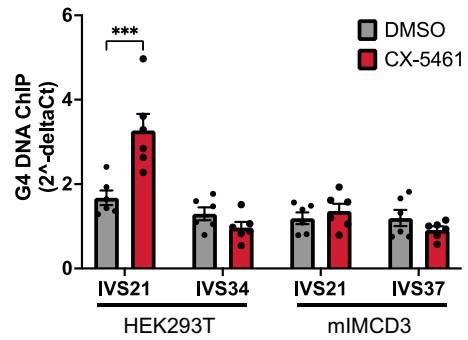

**Fig. 4 | G4 formation in h*PKD1*.** BG4-ChIP of HEK293T or mIMCD3 chromatin from cells treated with vehicle (DMSO), 10 μM Phen-DC3 (left) or 0.1 μM CX-5461 (right). Primers specific for a region adjacent to human *PKD1* IVS21 (G4-rich) or IVS34 (G4-poor), and mouse IVS21 or IVS37 (both G4-poor) were used in qPCR to determine enrichment. Locus amplification is displayed as $2^{-\text{delta Ct}}$. For Phen-DC3, data are presented as mean values ± s.e.m. (*n* = 6 from two independent experiments)

*** = *P* < 0.001, *P* > 0.999 for IVS34, IVS21 (mouse), and IVS37. The experiment was repeated with another G4-ligand, CX-5461 (right), and data are presented as mean values ± s.e.m. (*n* = 6 from two independent experiments) *** = *P* < 0.001, *P* = 0.681 for IVS34, *P* = 0.959 for IVS21 (mouse), and *P* = 0.785 for IVS37. Source data provided as a Source Data file.

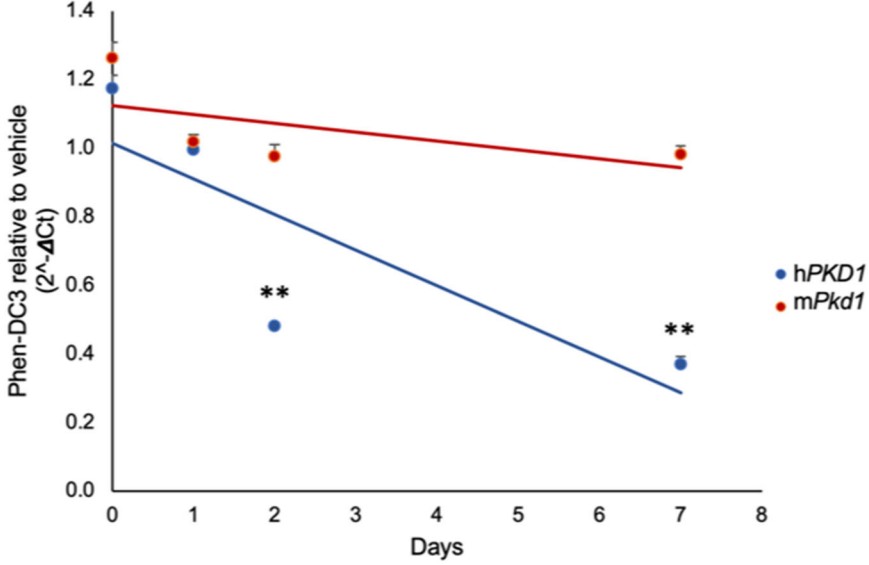

**Fig. 5 | G4 DNA impacts the expression of h*PKD1*.** qPCR of *PKD1* mRNA from HEK293T or mIMCD3 cells incubated with 10 μM Phen-DC3 at indicated timepoints. cDNA abundance is relative to DMSO treatment. Data are presented as mean values ± s.e.m. (*n* = 3 independent experiments). RM one-way ANOVA, ** = *P* < 0.01.

For HEK293T, day 0–1 *P* = 0.042, day 0–2, *P* = 0.003, day 0–7 *P* = 0.002. For mIMCD3 day 0–1 *P* = 0.119, day 0–2 *P* = 0.099, day 0–7 *P* = 0.019. Source data provided as a Source Data file.

## Methods

### Ethics

Frozen non-fixed ADPKD and normal human kidney tissue were generous gifts from the University of Kansas Medical Center provided by the NIDDK sponsored (NIH DK126126) Polycystic Kidney Disease Research Resource Consortium. Tissue was de-identified and affirmed to be non-human subjects research by the University of Kansas Medical Center and Western Michigan University Homer Stryker MD School of Medicine Human Research Protection Programs.

### Sequences, programs, statistics

Predicted non-overlapping G4 DNA motifs for h*PKD1* and *mPkd1* were quantitated using QGRS mapper https://bioinformatics.ramapo.edu/QGRS/index.php[22] using a 45-nucleotide window, 3 tandem repeats minimum and 8 nucleotides gap between repeats. Returned G4 DNA motifs had a G-score[22] > 64. Both strands were queried between the start and stop codons of each gene. Experimentally detected G4 DNAs

were curated by Endoquad https://endoquad.chenzxlab.cn/#/group-g4[23]. Primers were designed with NCBI primer blast and synthesized by Integrated DNA technologies. Sequences for PCR are included in Supplementary Table 1, Supplementary Table 2, and Supplementary Table 3. Drawings were created using Biorender.com. Significance was calculated and graphs generated using Prism software. Unless otherwise stated, standard error is shown in graphs and significance was calculated from at least six technical replicates from two independent experiments.

### Reagents and cells

Phen-DC3 (Sigma, SML2298) and CX-5461 (MedChemExpress, HY-13323) were dissolved in DMSO and used at a working concentration of 10 μM and 100 nM, respectively. BG4 antibody was purchased from Millipore (MABE917), and anti-γH2AX and anti-RAD51 from Novus Biologicals (NB 100-74435, NB 100-148). PC-1 antibody (7E12) (1/200, Western) was graciously provided by Dr. Chris Ward,

a.

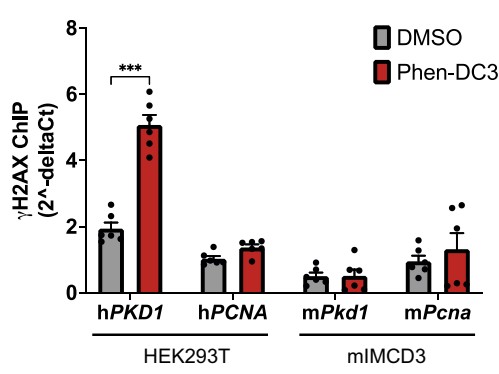

Phen-DC3 treated cells

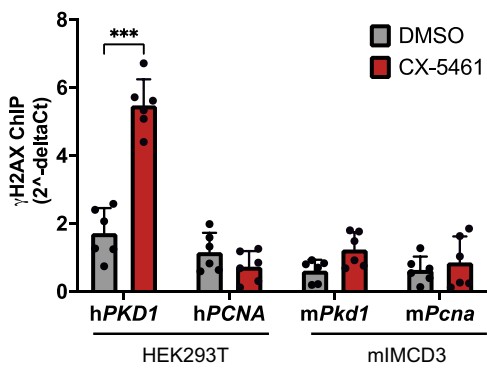

CX-5461 treated cells

b.

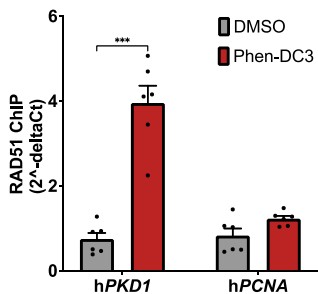

Phen-DC3 treated HEK293T

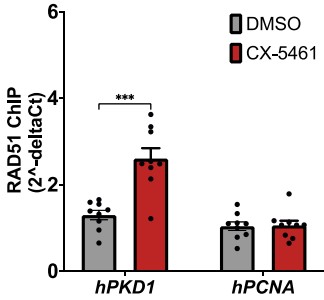

CX-5461 treated HEK293T

**Fig. 6 | G4 stabilization results in activation of the DNA damage response at *PKD1*. a** qPCR data for *PKD1* and *PCNA* from ChIPs of genomic DNA with anti-γH2AX antibody in DMSO, Phen-DC3 (left) or CX-5461-treated (right) HEK293T and mIMCD3 cells. For both ligands, data are presented as mean ± s.e.m. (*n* = 6 from two independent experiments for each ligand (four total experiments/IPs)), two-way ANOVA, *** = *P* < 0.001. For Phen-DC3, h*PCNA P* = 0.784, m*Pkd1 P* > 0.999, and m*Pcna P* = 0.741. For CX-5461, h*PCNA P* = 0.623, m*Pkd1 P* = 0.263, m*Pcna P* = 0.947.

**b** qPCR data for *PKD1* and *PCNA* from ChIPs of genomic DNA precipitated with anti-RAD51 antibody in DMSO, Phen-DC3 (left) or CX5461-treated (right) HEK293T. Data are presented as mean ± s.e.m. (*n* = 6 from two (Phen-DC3) and *n* = 9 from three (CX-5461) independent experiments), two-way ANOVA, *** = *P* < 0.001. For Phen-DC3 h*PCNA P* = 0.431, for CX-5461 h*PCNA P* = 0.931. Amplification results for ChIPs are displayed as 2^−delta Ct. Source data provided as a Source Data file.

University of Kansas Medical Center. SG4 and SG4-R105A plasmids (pHEN2-SG4 and pHEN2-SG4 R105A) were gifts from Dr. Shankar Balasubramanian[27](Addgene plasmids 196071 and 196072). SG4 nanobodies were purified essentially as described[27]. Briefly, BL21 (DE3) E. coli with each plasmid were induced at an $OD_{600}$ 0.4, with 0.5 mM IPTG, cells collected after overnight incubation at 28 °C and protein purified by Nickel chromatography. Protein was eluted with imidazole (200 mM) and dialyzed in PBS at 4 °C. Protein purity (>95%) was judged by SDS-PAGE and Coomassie staining. Proteins were brought to 5% glycerol and stored at −80 °C. HEK293T were a gift from Tom Rothstein (Western Michigan University Homer Stryker MD School of Medicine) and mIMCD3 was purchased from the ATCC (cat# CRL-2123). HEK293T cells were cultured in opti-MEM (Gibco, 31985-070) with 10% FBS and 1% penicillin-streptomycin (Corning, 30-001-CI); mIMCD3 cells were cultured in DMEM (Gibco, 11995-065) with 10% FBS and 1% penicillin-streptomycin.

**G4 DNA detection in vitro**

G4 DNA oligonucleotide for the dot blot (Fig. 2a) corresponds to a repeat in intron 1, 5′-TTTTTAGAGGTGGGAGGGGCTGGCAGGGA GGGAGAGGT, except for the additional 5′ thymines. The GT oligo, 5′-TTTTTAGAGGT**GT**GAG**TT**GCTGGCAG**T**GAG**T**GAGAGGT is the same as

the G4 DNA repeat except where guanines were interrupted with thymine (bolded). Each oligo was suspended in TE with 100 mM KCl and folded by incubation in a 98 °C water bath that was allowed to slowly come to room temperature. 200 pM of oligo, diluted 1:1 in the same buffer, was applied by a dot blot apparatus to Hybond nylon membrane (Amersham, PN 303 N), cross-linked by exposure to UV transilluminator for 2 min, and then blocked in 5% milk in TBS (50 mM Tris, 50 mM KCl), followed by incubation with SG4 or SG4-R105A (1/10 dilution) overnight, SG4-R105A is mutant for G4 binding[27]. Membranes were washed 3 times with TBS-tween, followed by 1 h incubation with 1/ 800 dilution of a rabbit anti-FLAG antibody (Cell Signaling Technology cat# 14793S). The membrane was then washed 3 times with TBS-T, followed by incubation for 1 h with HRP-conjugated anti-rabbit IgG (Thermo Scientific, 65-6120) diluted 1/3000 in TBS. Unbound antibody was removed with 3 washes of TBS-T and luminescence detected after 5 min incubation with WesternSure premium chemiluminescent substrate (Li-COR, 926-95000) using a Li-COR Western Blot imager. The same blot was stained with SYBR gold (Invitrogen, S11494) and imaged to show oligonucleotide loading. A representative dot blot is shown for no less than four independent experiments. CD spectra were collected as previously described[43]; briefly, the G4 oligonucleotide is derived from an IVS1 repeat 5′- CTGGCAGGGAGGGAGAGGTGGG

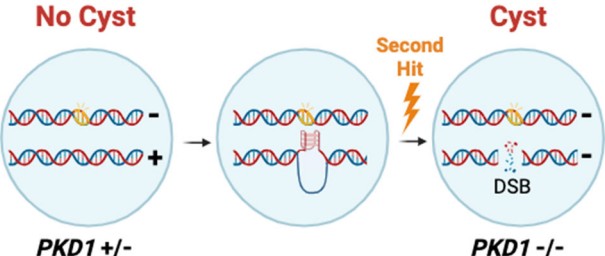

**Fig. 7 | Model for G4 DNA-induced second hit mutations in h*PKD1*.** Somatic cells heterozygous for a pathogenic *PKD1* allele (*PKD1*+/−) do not lead to cysts, left. G4 DNA forms in *PKD1* during replication, center. G4 DNAs block DNA metabolism and increases the risk of double strand breaks (DSB) in the remaining normal allele, right. Second hit inactivation (*PKD1*−/−) due to G4 DNA formation lowers polycystin-1 levels and leads to cell proliferation and cystogenesis. Created in BioRender, Bouma, J. (2024) BioRender.com/v35i306.

AGGGGCTGGCA, and the GT control contains 6 substitutions (underlined) to decrease G4 folding capability 5′- CTGGCAG**T**GAG**T**GA-GAGGA**G**TGAG**T**G**T**CTGGCA. Both oligonucleotides were subjected to G4 DNA folding conditions in a solution of TE containing 100 mM KCl incubated in a 98 °C water bath that was allowed to slowly come to room temperature. An Aviv model 215 spectrometer was used at 37 °C with a 1 cm path length cuvette. The spectra for parallel G4 structures peaks at 260, and dips at 240 nm[24,25]. CD results are shown for the average of three scans from one of two independent assays.

### *PKD1* RT-qPCR
HEK293T and mIMCD3 cells were incubated with DMSO alone or with 10 µM Phen-DC3 in DMSO. Cells were collected, washed with PBS, and mRNA prepared using a total RNA Miniprep kit (New England Biolabs, T2010S) at the indicated time points. For the zero timepoint, cells were collected after <1 h of DMSO or Phen-DC3 addition and RNA extracted using a Total RNA Miniprep kit (NEB, T2010S) and 500 ng was converted to cDNA with a Protoscript cDNA kit (New England Biolabs, E6560S). One microliter of cDNA was used in qPCR with primers specific to h*PKD1* and m*Pkd1*. The amount of *PKD1* amplification was normalized to β-actin qPCR. qPCR reactions were completed in triplicate using ThermoScientific QuantStudio qPCR machine. Reactions used 1x PowerUp™ SYBR™ green (Applied Biosystems, A25742). The amount of *PKD1* amplification was normalized to β-actin by subtracting to generate DeltaCt values, and amplification results were displayed as 2^-DeltaCt[44].

### Immunofluorescence and Immunohistochemistry
HEK293T were fixed with 100% ice cold methanol, washed 3 times in PBS, followed by incubation with PBS 1% Triton at 37 °C for 30 min for permeabilization. Cells were blocked with 10 % normal goat serum (Vector labs, S-1000), treated with 500 ng/µl RNase A (New England Biolabs, #T3018) and incubated overnight at 4 °C with BG4 (1:50) primary antibody. The following day cells were washed 3 times with PBS-T and incubated with 1:800 Rabbit anti-FLAG (Cell Signaling Technology, cat# 14793S) in PBS-T + 1% goat serum for 1 h at 37 °C. After washing 3 times with PBS-T, anti-rabbit Alexa Fluor 568 (Invitrogen, cat# A11011) 1:1000 was added and incubated 1 h at room temperature. After 3 washes with PBS, antifade mounting medium with DAPI (VECTA-SHIELD, H-1500) were added to mounted slides (blue). Images were collected using a Nikon A1R+ confocal microscope.

ADPKD tissue was taken from a minimally cystic region from one 56-year-old male with a BUN of 45 mg/dL and creatinine of 7.03 mg/dL. Normal human kidney tissue was derived from an aged-matched male. The tissue blocks were embedded in Tissue Tek Optimal Cutting Temperature compound (Sakura Finetek, 4583) and sectioned. Five micron-thick kidney sections were fixed by incubation in ice-cold methanol for 10 min at −20 °C, then washed in PBS, blocked with 10% normal goat serum (NGS) for 1 h at room temperature, then treated with SG4 or control (SG4-R105A)[27] primary antibodies (1:50) overnight at 4 °C. The following day tissue sections were washed three times with PBS at RT and incubated with rabbit anti-FLAG (1:800, Cell Signaling Technology, cat# 14793S) secondary antibody for 1 h at RT. Tissue sections were washed three times in PBS at RT and incubated with FITC goat anti-rabbit (1:400, Vector FL-1000) for 1 h at RT. Following washing three times in PBS, sections were mounted with VECTASHIELD antifade with DAPI and slides were viewed on a Zeiss Axioskop fluorescence microscope and images captured with a SPOT RT sCMOS digital camera.

CASFISH imaging followed methods described by Deng et al.[28]. with the following modifications; SNAP-tagged dCAS9, TMR-snap ligand, and sgRNA synthesis kit (New England Biolabs, #M0652, #S9105S, #E3322V) were used as reagents, and since tween disrupted tissue morphology PBS alone was use in washes. 94 sgRNAs were designed using ChopChop, https://chopchop.cbu.uib.no/ and targeted to a ~12 kb region encompassing the 3′ end of *PKD1* starting at exon 35 and extending into the 3′ tail of adjoining *TSC2*, with the intent to limit sgRNA reactivity with *PKD1* pseudogenes. Oligonucleotides were o-pool DNAs generated by Integrated DNA technologies. Sequences for sgRNA synthesis are available in Supplementary Data 1. Following labeling of normal human or ADPKD kidney tissue sections with SG4 or R105, as described above, sections were incubated with Oregon Green (NEB, cat# S9104S) or TMR-STAR (NEB, cat# 9105S)-labeled dCAS9 assembled with h*PKD1* sgRNAs, described above, at 37 °C for 30 min. Following washing three times with PBS, sections were mounted with Vectashield medium with DAPI (Vector) and images captured using a Nikon A1R+ confocal microscope. For controls, sections were also incubated with SG4 nanobody, followed by Fluorescein or Texas Red labeled dCAS9, without h*PKD1* sgRNAs.

### Statistics and reproducibility
Images selected for normal human or ADPKD tissue labeling were representative and from one of four independent labeling experiments. For qPCR of cDNA, results were from three independent experiments and RM one-way ANOVA was used to calculate significance. ChIP experiments were repeatable, and data is shown for two or more independent experiments with three technical replicates each. Assays for each ChIP experimental endpoint (i.e., G4 formation (Fig. 4) or DNA damage response (Fig. 6)) are represented by four independent experiments using two different G4-specific ligands (Phen-DC3 and CX-5461). Significance was calculated with Prism using two-way ANOVA.

### BG4 IP and chromatin IPs
HEK293T or mIMCD3 cells were plated at a density of $1.5 \times 10^6$ cells/plate and allowed to come to 90% confluence prior to treatment with 10 µM Phen-DC3, 100 nM CX-5461, or DMSO alone for 5 h, then cross-linked with 1% formaldehyde for 10 min. Crosslinking was halted by addition of 125 mM glycine. PBS-washed cells were collected by centrifugation and the cell pellet resuspended in lysis buffer (50 mM HEPES, 140 mM NaCl, 1 mM EDTA, 1% triton X-100, 0.1% sodium deoxycholate, 0.1% SDS, and protease inhibitor) then sonicated to produce fragments averaging 1–2 kb. BG4 antibody (1/40), anti-γH2AX (1/100), and anti-RAD51 (1/100) were used for IP experiments where indicated. Anti-FLAG magnetic beads (Sigma, cat# M8823) washed with calf thymus DNA were used to IP BG4-associated DNAs, and protein A resin beads (GenScript, L00210S) were used to IP γH2AX-associated DNAs. Crosslinks from precipitated chromatin were reversed at 65 °C overnight. DNA was purified with Monarch PCR & DNA cleanup kit (New England Biolabs, T1030S). DNA enrichment was quantified by qPCR using h*PKD1* or m*Pkd1*-specific and h*PCNA* or m*Pcna*-specific primers. PCR primers were designed to amplify a

region 267 nt upstream of a G4 DNA-dense (IVS21) intron and near a G4 DNA-sparse (IVS34) intron within h*PKD1*, allowing comparisons of relative enrichments for two regions within the same gene. For m*Pkd1*, primer sets were selected to capture template enrichments of comparable nucleotide positions (IVS21 and IVS37). Amplicons were verified by agarose electrophoresis and DNA sequencing. Input DNAs were equal and verified by qPCR. Following real time PCR, raw target Ct values were collected from chromatin incubated with beads only (no antibody, background) as well as chromatin incubated with beads plus primary antibody. Data was normalized (delta Ct) by subtracting the raw Ct value of beads only background from the raw Ct value of beads and antibody IP (i.e., $Ct_{antibody+beads} - Ct_{beads}$). Results displayed are from two or more independent assays and are representative of no less than four independent experiments.

## Reporting summary

Further information on research design is available in the Nature Portfolio Reporting Summary linked to this article.

## Data availability

Data is available within the paper, supplementary figures, and the source data file. Source data are provided with this paper.

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

## Acknowledgements

Research was supported by NIH/NIDDK R15DK119864 to E.L. and by Western Michigan University Homer Stryker MD School of Medicine. Studies utilized resources provided by the Kansas PKD center (NIH DK126126) and NIDDK sponsored Polycystic Kidney Disease Research Resource Consortium. The authors wish to thank Kristi Bailey, WMed histology, for assistance with tissue sectioning and Michael Clemente, WMed Imaging Core, for assistance with confocal microscopy. Drawings were created with BioRender.com.

## Author contributions

E.L. directed and supervised the research. E.L., G.J.B, and G.B.V.H conceived experiments and edited the manuscript. J.D. and A.L.Z. analyzed G4 sequences in *PKD1*. G.A. performed CD scans. A.M.P performed ChIPs, dot blot, Westerns, qPCR, cell culture, and created diagrams with G.J.B. cDNA analysis was performed by A.M.P and S.B., G.B.V.H and J.K. performed IF microscopy and CASFISH. All authors reviewed, provided comments, and approved the manuscript.

## Competing interests

A pending U.S. Provisional Patent Application No. 63/693,922 (inventor E.L.) directed to G4 quadruplex structures in the *PKD1* gene has been filed on behalf of the applicant, Western Michigan University Homer Stryker M.D. School of Medicine. The authors declare no other competing interests.
