## [Peer Review File · Nature Communications]

REVIEWER COMMENTS

Reviewer #1 (Remarks to the Author):

The article “guanine quadruplex DNA provokes DNA breaks for second-hit mutagenesis of human PKD1” by Parsons and colleagues studies a very timely and interesting topic.

The aim of the study was to characterize the contribution of a DNA G4 structures within the tumor suppressor PKD1 in humans for its relevance to obtain second-hit mutations that support tumorigenesis. Currently it is not known why some sequence regions are more prone for second-hit mutations than others and why tumor suppressors or among those that often obtain those. In particular human PKD1 is highly mutated which may be due to an intronic repeat region, which can form G4s. Although the topic is very interesting the manuscript and conclusions are at an early step and multiple in cell experiments are required to support their claim. In particular are G4s forming at these regions during second-hit, or are mutations accumulating at G4 forming regions in patients/samples after a second-hit? Further also it is required to address why a G4 will form at these regions, if in their settings it only forms with G4 stabilizing drugs?

See major specific concerns regarding the manuscript which I have listed below:

Figure 1:

Because mouse PKD1 has not the strong mutagenetic nature they hypothesized that loss in G4 forming potential are causing the reduced mutational rates in mouse. Figure 1 shows in silico analysis of PKD1 in mouse, rat and human. With the conclusion that mouse PKD1 has less potential to form G4s. Most of the figure is in silico and it is needed to know the search criteria, what kind of G4 motifs were detected? Stability (G-score) how much do they overlap in particular at their region of interest? Why are in Figure 1B 124 G4 motifs identified and in the mapper around 60 annotated? The experiment in Figure 1C are not clear to me, are here oligos used and folded into a G4 and G4 formation tested by Dotblot? Here information on why those were hand-selected and also why are not standard in vitro G4 detection methods used (e.g. CD, melting experiments, in-gel stainings). What are the four spots?

Figure 2:

Here they provided evidence that HEK and tissue of ADPKD can form G4s. They used for HEK cells BG4 and for the tissue SG4. Please explain why different staining methods were used. Here quantification and comparison of stainings is needed. So far in the literature all cells have the ability to form G4s. The ChIP with BG4 is interesting, but I did not find information what the control region is? It is not indicated on Figure 1 (distance to IVS21, G4 content)

Figure 3, they conclude based on PhenDC addition that G4 stabilization acts in cis on PKD1 expression, as transcription factors were excluded. Here it needs to be shown that transcription factors are expressed still similar after G4 stabilization further, it also needs to be shown that PhenDC in general acts similar in mouse cells than in human cells. Meaning are they similar toxic to the cells? do they elevate similar globally G4 levels in the cells? it is known that globally gene expression is different and it is expected that mouse and human respond different this needs to be studied and discussed. Also the conclusion as γ H2A is elevated after DNA damage at this loci is interesting but too broad and here more work is needed. why are those sites not repaired? How is the damage created, if as they speculate in the abstract it is via replication this needs to be shown. Also mutation studies at that site are required to pinpoint everything to G4s.

Reviewer #2 (Remarks to the Author):

The authors investigated the involvement G4 structures on the inactivation of the PDK1 gene, which drives polycystic kidney disease. They performed a bioinformatic analysis to show that there is a significantly greater potential for G4-formation in the human PDK1 gene (hPDK1) relative to the mouse (mPkd1) and rat (rPDK1) PDK1 genes. This inferred difference in G4-abundance between murine and human PKD1 is significant as it may explain why mice with heterozygous Pkd1 mutations do not recapitulate the second-hit inactivation of PKD1 observed in humans. The authors demonstrate the environment of kidney cells is supportive of G4-formation by detecting G4s in renal cells and tissues from polycystic kidney disease patients using ChIP and microscopy experiments with G4-specific antibodies/nanobodies. Also noteworthy are the findings that G4-stabilization by G4-ligands downregulates PKD1 transcription and promotes the accumulation of DNA breaks in the hPDK1 gene. Altogether, the experiments in this manuscript support a model in which G4-formation is involved in the second-hit inactivation of hPDK1, which leads to polycystic kidney disease development in people harboring heterozygous PDK1 mutations. Broadly, the findings of this manuscript are significant to an emerging field of study focused on the health effects of G4-associated genomic instability because they implicate G4s in the inactivation of genes linked to disease and oncogenesis by the second-hit pathway. Additionally, this report implicates G4s in a disease other than cancer, where most current efforts are focused. Thus, the results open a new possibility of targeting G4s in other diseases as a potential therapeutic approach. The findings will be of interest to the readership of Nature Communications and are likely to have a significant impact, and publication is recommended if the following concerns can be addressed:

Major points:

1. Rider et al., 2022, JBC (citation 14 in the submitted manuscript) showed that a DNA sequence from the hPKD1 gene that forms G4s in vitro induces mutagenesis in cell lines in a replication-dependent manner, so a connection between PDK1 and G4-induced genomic instability already existed prior to the submission of the reviewed manuscript. The authors do present novel findings including that G4s can be detected in human renal cells and in tissue samples from polycystic kidney disease patients in ChIP and microscopy experiments. However, a key question remains unaddressed: do these effects actually lead to inactivating second hit mutations in the PKD1 gene? Experiments are needed to demonstrate this. If no tractable models for mutagenesis of the PKD1 gene are available, an alternative model system that is prone to forming G4 structures could be used to demonstrate the link.
2. Another novel finding is that G4-stabilization regulates transcription of PKD1, supporting G4s do form in hPKD1 in cells and suggesting G4-formation can silence PKD1 by an additional mechanism other than driving mutagenesis of PKD1. The transcriptional regulation of PKD1 by G4-stabilization also implies a functional role may exist for G4s in the modulation of PDK1 activity in healthy human cells. Microscopy work showing G4-formation in tissues from polycystic disease patients supports G4-induced genomic instability is a possibility in polycystic disease development. However, the depletion of PKD1 should be confirmed at the protein level, and discussion is needed regarding whether G4-dependent suppression of transcription is expected to be a physiologically relevant mechanism underlying the etiology of polycystic kidney disease.
3. The work mostly supports the conclusions and claims, but the claim that renal cells are in an environment that can support G4-formation would be made stronger via detection of G4s in renal cells and patient samples with additional G4 antibodies. The authors should consider performing microscopy experiments with an additional G4 antibody/nanobody/light-up probe other than SG4 (such as 1H6) because SG4 clearly has background binding to non-G4 capable sequences in the dot blot. Imaging the polycystic kidney disease tissues with an additional G4-antibody will also strengthen the argument that G4-formation does occur in human kidneys. Performing the dot blot with another G4-antibody with the G4 and GT oligos should accompany these experiments.
4. Further discussion is needed regarding potential genetic evidence that might support the proposed mechanism whereby G4-associated genome instability promotes diseases that result from gene inactivation. For example, are individuals with mutations in genes like BLM that are involved in the resolution of G4 structures more susceptible to polycystic kidney disease? Can G4-associated instability be linked to loss of heterozygosity more generally using publicly available datasets or previously published analyses?
5. Mycoplasma testing should be done for HEK293T cells to be sure findings (especially PDK1 gene expression data in Figure 3A) are not artifacts of contamination. If mycoplasma contamination is found experiments should be repeated in mycoplasma-free cells to see if similar results are produced. Regardless, confirming these results in at least one more cell line would increase the robustness of the analysis.

Minor Points:

1. While the dot blot with SG4 in Figure 1d does suggest that the G4 oligo derived from hPKD1 forms G4 in vitro, results should be included from another method to show in vitro G4-formation of the oligo, especially since the SG4 antibody does show some background binding to the GT control. Polymerase stop assays or circular dichroism for the G4 and GT oligos would be helpful.
2. Some explanation is needed as to why tween detergent was not added to TBS when the SG4 dot blots were washed after antibody treatments. Doing so may help reduce the non-specific binding observed by SG4 in the dot blot. A similar question arises for the SG4-treated patient tissues. Discussion is needed since significant background staining of the control GT oligo with SG4 was observed in the dot blot.
3. The methods section is commendably detailed in most respects, but some additional details are needed. One thing missing is the dilution of primary BG4 antibody that was used in immunofluorescence experiments done with the HEK293T cells under the “Immunofluorescence and Immunohistochemistry” section of the methods. Additionally, while the authors cite reference 28 in the calculation of relative gene expression and protein/chromatin enrichment values via the comparative CT method represented as $2^{-\Delta\Delta CT}$ for the results in Figures 2B, 3A, 3B and Supplementary Figure 1, since there can be multiple ways to calculate this, the exact equations used should be provided.
4. Reference 25 (line 301) is missing a “T” in the title of the manuscript.

Reviewer #3 (Remarks to the Author):

I co-reviewed this manuscript with one of the reviewers who provided the listed reports. This is part of the Nature Communications initiative to facilitate training in peer review and to provide appropriate recognition for Early Career Researchers who co-review manuscripts

Reviewer #4 (Remarks to the Author):

This is an original work by Parsons et.al. that provides a potential mechanism behind the second hit theory which causes inactivation of tumor suppressors, specifically focusing on human PKD1 gene as a model system. ADPKD is a monogenic disease caused by germline mutations in PKD1 or PKD2.

As opposed to humans where the germline pathogenic allele is joined by somatic inactivation of the remaining functional allele (second hit), heterozygous mice are protected from this somatic allele inactivation. The study provides strong evidence that guanine quadruplex (G4) DNA structures in humans promote DNA breaks which causes mutagenesis of the somatic DNA strand in PKD1 leading the autosomal polycystic kidney disease. These mechanisms are low/absent in mice. While the premise of the study is very strong, and the data presented is exciting and leading to the appropriate conclusions. There are several weaknesses that require attention.

1. Figure.1a. not clear what exactly the figure shows, please label the G4 structure properly showing G4 motifs.
2. Fig;1d. No indication how many times the dot blot was repeated, whether the results are significant or not. The difference between G4 and GT in SG4 panel does not seem much. Dot intensity in at least three blots should be presented with quantification of the band intensity normalized against SYBR.
3. It is not clear whether statistical analysis was performed for all the quantifiable figures. If so, the significance should be indicated in the bar graph as p values/asterisks.
4. Figure2a. Immunolabeling of G4 quadruplex was done in HEK293T and human ADPKD tissue, what was the rationale behind that? Comparisons between normal human kidney and ADPKD kidney and cells from normal human and ADPKD patients would make more sense.
5. In the text, page2, line63, the authors mention that Immunolabeling was performed to test if G4 folding was physiological relevant. How would labeling tell the physiological relevance specifically of PKD1 when BG4 labeled nuclei of both 293T cells and ADPKD tissue. Is G4 folding not expected to be present in all human cells? How would this confirm that renal microenvironment is permissive to genomic G4 formation, when no other tissue other than ADPKD kidney/293T cells were used.
6. Comparisons of G4 labeling on renal and extrarenal tissues of mice and human would make more sense to show that renal environment specific G4 formation.
7. Figure 2b: Please indicate significance by asterisks. Provide number of replicates in the figure legends and preferably provide bar graphs with individual dots for each sample used.
8. What would be the off-target effects of G4 targeting therapies, should be discussed.
9. Figure 3: n=3 replicates from two experiments for these studies are mentioned in material and methods appear to be low to show significance.
10. Page 4, line 154 and 156 and , IVS1 is written instead of IVS21.

Modifications to the manuscript are indicated with **red text**. Specific responses are listed below for each reviewer comment, in *red italics*.

REVIEWER COMMENTS

Reviewer #1 (Remarks to the Author):

The article “guanine quadruplex DNA provokes DNA breaks for second-hit mutagenesis of human PKD1” by Parsons and colleagues studies a very timely and interesting topic.

The aim of the study was to characterize the contribution of a DNA G4 structures within the tumor suppressor PKD1 in humans for its relevance to obtain second-hit mutations that support tumorigenesis. Currently it is not known why some sequence regions are more prone for second-hit mutations than others and why tumor suppressors or among those that often obtain those. In particular human PKD1 is highly mutated which may be due to an intronic repeat region, which can form G4s. Although the topic is very interesting the manuscript and conclusions are at an early step and multiple in cell experiments are required to support their claim. In particular are G4s forming at these regions during second-hit, or are mutations accumulating at G4 forming regions in patients/samples after a second-hit? Further also it is required to address why a G4 will form at those regions, if in their settings it only forms with G4 stabilizing drugs?

Thank you for the thoughtful review, we hope the addition of seven new experiments allay concerns that the research is at an early stage and more experiments are needed. We have made clarifications in the text, emphasized prior research on PKD1 repeats, added paragraphs discussing G4 DNA and mutagenesis, expanded on prior research on G4 DNA and PKD1 mutations, and discussed the potential regulatory roles of G4 DNA in PC-1 expression. We also adjusted the title to better reflect the conclusions.

See major specific concerns regarding the manuscript which I have listed below

Figure 1:

Because mouse PKD1 has not the strong mutagenetic nature they hypothesized that loss in G4 forming potential are causing the reduced mutational rates in mouse. Figure 1 shows in silico analysis of PKD1 in mouse, rat and human. With the conclusion that mouse PKD1 has less potential to form G4s. Most of the figure is in silico and it is needed to know the search criteria, what kind of G4 motifs were detected? Stability (G-score) how much do they overlap in particular at their region of interest?

The G4 search criteria is explained in the methods section, line 191. The relationship between human PKD1 and mouse Pkd1 G4 content remains the same independent of the search criteria used, so we make a point of also citing Endoquad, which shows the same relationship; G4 is abundant in human but not mouse Pkd1. The G-score is on line 193.

Why are in Figure 1B 124 G4 motifs identified and in the mapper around 60 annotated?

We are unsure about the confusion, 124 G4 motifs are annotated on the map of the human gene in 1b. Each motif is annotated with a + symbol, described in the figure legend.

The experiment in Figure 1C are not clear to me, are here oligos used and folded into a G4 and G4 formation tested by Dotblot? Here information on why those were hand-selected and also why are not standard in vitro G4 detection methods used (e.g. CD, melting experiments, in-gel stainings). What are

the four spots? *We updated the figure with DNA concentrations, provided G4-folding conditions in the methods, and repeated the dot blot with better wash conditions to reduce non-specific binding. The G4 motif selected is representative of a G-rich (poly-purine) repeat, but it found in intron 1 instead of intron 21. CD was added further showing G4 formation. G4 folding for a PKD1 poly-purine repeat was previous by Rider et al., and so we cited those results in multiple places throughout the manuscript. Thus, G4 formation has now been described for intron 1 repeats and intron 21 repeats (both are similar poly-purine repeats).*

Figure 2:

Here they provided evidence that HEK and tissue of ADPKD can form G4s. They used for HEK cells BG4 and for the tissue SG4. Please explain why different staining methods were used. Here quantification and comparison of stainings is needed. So far in the literature all cells have the ability to form G4s. The ChIP with BG4 is interesting, but I did not find information what the control region is? Its not indicated on Figure 1 (distance to IVS21, G4 content)

We agree, the IF results were fully expected as all cells appear to form G4, so we added this statement on line 87 and removed "renal cell environment" and "permissive" to reduce confusion. Those images were also moved to Supl. Fig 1, since they just show that BG4 recognizes G4 DNA. The new image, fig. 2, shows co-localization of PKD1 with G4, and this data better supports our conclusions.

For BG4-ChIPs, IVS34 is the control, which is a G4-poor region within PKD1. Thus, we compare a G4-rich region to a G4-poor region. We added a sentence to clarify this on line 92-93: "Exon/Intron 34 has 2 potential G4 motifs compared to the 38 in intron 21 (Fig. 2c), making it a good negative control.". Relevant introns are marked in fig 1c to improve clarity.

Figure 3, they conclude based on PhenDC addition that G4 stabilization acts in cis on PKD1 expression, as transcription factors were excluded. Here it needs to be shown that transcription factors are expressed still similar after G4 stabilization further, it also needs to be shown that PhenDC in general acts similar in mouse cells than in human cells. Meaning are they similar toxic to the cells? do they elevate similar globally G4 levels in the cells? it is known that globally gene expression is different and it is expected that mouse and human respond different this needs to be studied and discussed.

The overall concern appears to be whether G4 affects PKD1 in cis or trans. Unfortunately, PKD1 transcription factors are unknown so further study on them is not possible yet. Since G4 stabilization reduces hPKD1 but not mPKd1, and only the former gene encodes abundant G4, our cautious interpretation here is reasonable as we avoid over-stating the relevance of Phen-DC3 effects. Certainly, our results open a new area of research; defining the roles of naturally formed G4 DNAs in PKD1. We re-wrote this section to emphasize that trans-acting effects cannot be excluded, and that more study is needed to define the mechanisms of gene regulation, starting on line 109. A new paragraph is also included discussing models for G4-based gene regulation, starting on line 137.

Also the conclusion as γ H2A is elevated after DNA damage at this loci is interesting but too broad and here more work is needed. why are those sites not repaired? How is the damage created, if as they speculate in the abstract it is via replication this needs to be shown. Also mutation studies at that site are required to pin point everything to G4s.

We agree, the mechanisms of break formation in PKD1 are an important area of future research. The conclusion we draw from the results is that DNA breaks form in PKD1 because of G4 formation. Since breaks are known to be genotoxic, and precursors to PKD1 inactivation, we added those clarifications, see line 133-135.

For the second concern, γ H2AX signals the DNA damage response. It is commonly used marker for DNA damage, originally described in 1998 by Rogakou et al.. DNA breaks are genotoxic, and indeed this is the principle behind general chemotherapies and radiation treatment for cancer, which induce DNA breaks to kill proliferating cells. We provided more supporting data, including a ChIP experiment with anti-RAD51 (a recombination repair protein) to directly show that recombination repair factors are recruited to hPKD1 in response to G4-stabilization, one of many likely repair responses. We also mention that G4-induced DNA breaks lead to LOH or gene conversion on lines 133-135, both of which are connected with PKD1 second hits. Finally, we added a discussion on the results of Rider et al., which used a portion of IVS21 to show mutagenesis by G4-dependent break induced replication on lines 154-156. Together, we hope that better connects the dots for how G4 DNA provokes PKD1 mutagenesis.

Reviewer #2 (Remarks to the Author):

The authors investigated the involvement G4 structures on the inactivation of the PDK1 gene, which drives polycystic kidney disease. They performed a bioinformatic analysis to show that there is a significantly greater potential for G4-formation in the human PDK1 gene (hPKD1) relative to the mouse (mPkd1) and rat (rPDK1) PDK1 genes. This inferred difference in G4-abundance between murine and human PKD1 is significant as it may explain why mice with heterozygous Pkd1 mutations do not recapitulate the second-hit inactivation of PKD1 observed in humans. The authors demonstrate the environment of kidney cells is supportive of G4-formation by detecting G4s in renal cells and tissues from polycystic kidney disease patients using ChIP and microscopy experiments with G4-specific antibodies/nanobodies. Also noteworthy are the findings that G4-stabilization by G4-ligands downregulates PKD1 transcription and promotes the accumulation of DNA breaks in the hPKD1 gene. Altogether, the experiments in this manuscript support a model in which G4-formation is involved in the second-hit inactivation of hPKD1, which leads to polycystic kidney disease development in people harboring heterozygous PDK1 mutations. Broadly, the findings of this manuscript are significant to an emerging field of study focused on the health effects of G4-associated genomic instability because they implicate G4s in the inactivation of genes linked to disease and oncogenesis by the second-hit pathway. Additionally, this report implicates G4s in a disease other than cancer, where most current efforts are focused. Thus, the results open a new possibility of targeting G4s in other diseases as a potential therapeutic approach. The findings will be of interest to the readership of Nature Communications and are likely to have a significant impact, and publication is recommended if the following concerns can be addressed:

Thank you for recognizing the significance of our results and offering helpful suggestions. We hope the addition of new experiments showing an improved dot blot, circular dichroism, co-localization of G4 DNA with the PKD1 locus, and an additional ChIPs showing G4-dependent RAD51 recruitment to hPKD1 strengthens the manuscript even more.

Major points:

1. Rider et al., 2022, JBC (citation 14 in the submitted manuscript) showed that a DNA sequence from the hPKD1 gene that forms G4s in vitro induces mutagenesis in cell lines in a replication-dependent manner, so a connection between PDK1 and G4-induced genomic instability already existed prior to the submission of the reviewed manuscript. The authors do present novel findings including that G4s can be detected in human renal cells and in tissue samples from polycystic kidney disease patients in ChIP and

microscopy experiments. However, a key question remains unaddressed: do these effects actually lead to inactivating second hit mutations in the PKD1 gene? Experiments are needed to demonstrate this. If no tractable models for mutagenesis of the PKD1 gene are available, an alternative model system that is prone to forming G4 structures could be used to demonstrate the link.

This is an excellent point. We adjusted the text in multiple places to emphasize the break-induced repair assay with an IVS21 repeat, showing hypermutation at that site due to G4 DNA (Rider et al., 2022). Since that was a tractable model for G4-induced mutagenesis, a better discussion of how our results fit into this prior research will hopefully strengthen the case that G4 DNA in PKD1 promotes DNA breaks, which lead to mutagenesis. The point is well taken that the next step needs to be experiments showing second hit inactivation, so we changed the title to better match the conclusions we draw from the results and added a paragraph discussing the model for G4 DNA in second hit mutagenesis on lines 151-168. We also added RAD51 ChIPs (Figure 3) and discussed how G4-induced DNA breaks could explain LOH and gene conversion in PKD1 second hits, see lines 133-135.

2. Another novel finding is that G4-stabilization regulates transcription of PKD1, supporting G4s do form in hPKD1 in cells and suggesting G4-formation can silence PKD1 by an additional mechanism other than driving mutagenesis of PKD1. The transcriptional regulation of PKD1 by G4-stabilization also implies a functional role may exist for G4s in the modulation of PDK1 activity in healthy human cells. Microscopy work showing G4-formation in tissues from polycystic disease patients supports G4-induced genomic instability is a possibility in polycystic disease development. However, the depletion of PKD1 should be confirmed at the protein level, and discussion is needed regarding whether G4-dependent suppression of transcription is expected to be a physiologically relevant mechanism underlying the etiology of polycystic kidney disease.

We wholeheartedly agree on the likely importance of G4 DNA to PKD1 expression, particularly as it relates to ADPKD. We anticipate our results will open this new area of investigation. While Phen-DC3 lowered PKD1 mRNA, caution is in order as natural G4 formation may have either positive or negative influences on mRNA production. We now make that clarification on lines 111-115. Also, a paragraph was added starting on line 137 to expand on this and highlight the need for future research on polycystin-1 expression. A western showing polycystin-1 protein after PhenDC3 exposure is added, supplementary figure 2.

3. The work mostly supports the conclusions and claims, but the claim that renal cells are in an environment that can support G4-formation would be made stronger via detection of G4s in renal cells and patient samples with additional G4 antibodies. The authors should consider performing microscopy experiments with an additional G4 antibody/nanobody/light-up probe other than SG4 (such as 1H6) because SG4 clearly has background binding to non-G4 capable sequences in the dot blot.

Thank you for this comment, we have added new experiments and adjusted the text to address the concern over SG4 specificity, and G4 detection in vitro. SG4 specificity concerns were addressed by adjusting the wash buffer for the dot blot, as suggested, which greatly reduced non-specific binding to the GT oligo (essential zero now). The background in the previous dot blot was most likely due to weak G4 formation, since G doublets were retained in the GT oligo. We also added a sentence about prior BG4 and SG4 characterization for specificity on lines 73-75. The ADPKD IF microscopy used BG4 not SG4, corrected now, and this was moved to Suppl. Fig. 1, and we include PKD1-G4 colocalization in a new Fig. 2 with normal and ADPKD tissue.

Imaging the polycystic kidney disease tissues with an additional G4-antibody will also strengthen the argument that G4-formation does occur in human kidneys. Performing the dot blot with another G4-antibody with the G4 and GT oligos should accompany these experiments.

We did not intend to imply that renal cells are unique in their ability to support G4 structures. IF results were anticipated. The text was adjusted to better explain that experiment, which was to simply test the BG4 antibody and rationalize the ChIPs. Since it is not vital to the manuscript, we moved that to Suppl. Fig. 1 and replaced with new IF microscopy showing PKD1-G4 co-localization, Fig. 2.

4. Further discussion is needed regarding potential genetic evidence that might support the proposed mechanism whereby G4-associated genome instability promotes diseases that result from gene inactivation. For example, are individuals with mutations in genes like BLM that are involved in the resolution of G4 structures more susceptible to polycystic kidney disease? Can G4-associated instability be linked to loss of heterozygosity more generally using publicly available datasets or previously published analyses?

We agree more discussion here will improve the manuscript. To our knowledge, we are the first to show that G4 DNAs are forming at the human PKD1 locus to cause DNA breaks. We added a new paragraph on G4-based gene regulation, and how that may apply to polycystin-1. Following that, we expanded on prior studies from Rider et al., and how those results on IVS21 G4s support our findings and then included a statement on BLM and FANCI deficiencies, however a connection to ADPKD is not obvious at this point since these patients suffer early cancers. Hopefully this better places our findings in the context of what is known about G4 DNA, second hit, and disease.

5. Mycoplasma testing should be done for HEK293T cells to be sure findings (especially PDK1 gene expression data in Figure 3A) are not artifacts of contamination. If mycoplasma contamination is found experiments should be repeated in mycoplasma-free cells to see if similar results are produced. Regardless, confirming these results in at least one more cell line would increase the robustness of the analysis.

All cells tested negative for mycoplasma.

Minor Points:

1. While the dot blot with SG4 in Figure 1d does suggest that the G4 oligo derived from hPKD1 forms G4 in vitro, results should be included from another method to show in vitro G4-formation of the oligo, especially since the SG4 antibody does show some background binding to the GT control. Polymerase stop assays or circular dichroism for the G4 and GT oligos would be helpful.

Circular Dichroism of the IVS1 G4 motif and GT control is now included in Fig. 1e

2. Some explanation is needed as to why tween detergent was not added to TBS when the SG4 dot blots were washed after antibody treatments. Doing so may help reduce the non-specific binding observed by SG4 in the dot blot. A similar question arises for the SG4-treated patient tissues. Discussion is needed since significant background staining of the control GT oligo with SG4 was observed in the dot blot.

Thank you for the excellent suggestion, we repeated dot blots with TBS-tween and non-specific binding was greatly reduced! We hope this addresses concern on SG4 specificity.

3. The methods section is commendably detailed in most respects, but some additional details are needed. One thing missing is the dilution of primary BG4 antibody that was used in immunofluorescence

experiments done with the HEK293T cells under the “Immunofluorescence and Immunohistochemistry” section of the methods.

Corrected, line 262.

Additionally, while the authors cite reference 28 in the calculation of relative gene expression and protein/chromatin enrichment values via the comparative CT method represented as $2^{-\Delta\Delta CT}$ for the results in Figures 2B, 3A, 3B and Supplementary Figure 1, since there can be multiple ways to calculate this, the exact equations used should be provided.

We added clarification to the methods: “The amount of PKD1 amplification was normalized to b-actin by subtracting to generate DeltaCt values, and amplification results were displayed as $2^{-\Delta\Delta Ct^{44}}$ ”

4. Reference 25 (line 301) is missing a “T” in the title of the manuscript.

Corrected.

Reviewer #3 (Remarks to the Author):

I co-reviewed this manuscript with one of the reviewers who provided the listed reports. This is part of the Nature Communications initiative to facilitate training in peer review and to provide appropriate recognition for Early Career Researchers who co-review manuscripts

Reviewer #4 (Remarks to the Author):

This is an original work by Parsons et.al. that provides a potential mechanism behind the second hit theory which causes inactivation of tumor suppressors, specifically focusing on human PKD1 gene as a model system. ADPKD is a monogenic disease caused by germline mutations in PKD1 or PKD2. As opposed to humans where the germline pathogenic allele is joined by somatic inactivation of the remaining functional allele (second hit), heterozygous mice are protected from this somatic allele inactivation. The study provides strong evidence that guanine quadruplex (G4) DNA structures in humans promote DNA breaks which causes mutagenesis of the somatic DNA strand in PKD1 leading the autosomal polycystic kidney disease. These mechanisms are low/absent in mice. While the premise of the study is very strong, and the data presented is exciting and leading to the appropriate conclusions. There are several weaknesses that require attention.

Thank you for the comments that our conclusions are appropriate and exciting. We have added new experiments to address weaknesses and hope that this revised manuscript is strengthened even more.

1. Figure.1a. not clear what exactly the figure shows, please label the G4 structure properly showing G4 motifs.

Fig. 1a has been edited to improve clarity. G's were added to the structure to indicate tetrads.

2. Fig;1d. No indication how many times the dot blot was repeated, whether the results are significant or not. The difference between G4 and GT in SG4 panel does not seem much. Dot intensity in at least three blots should be presented with quantification of the band intensity normalized against SYBR.

The dot blot was repeated, and a reviewer suggestion to include Tween in the wash greatly reduced non-specific binding to GT to essentially undetectable levels (Fig. 1d). This should address other concerns as

well (SG4 for IF microscopy). The number of repeats has been added to methods section. The dot blots are non-quantitative in nature and show the presence/absence of G4, so we included circular dichroism (Fig. 1e) on the intron 1 repeat and added a citation of previous G4 characterization for intron 21 DNAs (Rider et al.) as support for in vitro formation of G4 from PKD1 sequences.

3. It is not clear whether statistical analysis was performed for all the quantifiable figures. If so, the significance should be indicated in the bar graph as p values/asterisks.

Significance was calculated for all quantifiable figures, asterisks are now included to indicate significance and individual dots for each data point are now shown.

4. Figure 2a. Immunolabeling of G4 quadruplex was done in HEK293T and human ADPKD tissue, what was the rationale behind that? Comparisons between normal human kidney and ADPKD kidney and cells from normal human and ADPKD patients would make more sense.

We agree, the rationale was unclear. The result was fully expected and is not vital. That data was moved to suppl. Fig. 1 and replaced with the more important finding of G4-PKD1 co-localization in normal and ADPKD tissue, Fig. 2. We then follow those results with a more quantitative approach, ChIPs with BG4. We hope that is a more logical presentation of our findings.

5. In the text, page 2, line 63, the authors mention that Immunolabeling was performed to test if G4 folding was physiological relevant. How would labeling tell the physiological relevance specifically of PKD1 when BG4 labeled nuclei of both 293T cells and ADPKD tissue. Is G4 folding not expected to be present in all human cells? How would this confirm that renal microenvironment is permissive to genomic G4 formation, when no other tissue other than ADPKD kidney/293T cells were used.

We agree, these results are anticipated and did not mean to imply that renal cells have a special capacity to form G4. The word "permissive" was removed. A new IF image was added, Fig. 2 see above.

6. Comparisons of G4 labeling on renal and extrarenal tissues of mice and human would make more sense to show that renal environment specific G4 formation.

See above, renal cells are not unique in their ability to form G4 DNA. We added normal and ADPKD tissue to the SG4 immunofluorescence data in Fig. 2.

7. Figure 2b: Please indicate significance by asterisks. Provide number of replicates in the figure legends and preferably provide bar graphs with individual dots for each sample used.

Addressed, see #3 above. We describe the statistics and replicates for each experiment in the methods section.

8. What would be the off-target effects of G4 targeting therapies, should be discussed.

This is an excellent point. G4-stabilizing drugs for cancer result in mutagenesis (Koh et al., 2024), but destabilizing treatments are novel and essentially theoretical at this point. We added a paragraph better discussing G4 DNA and the implications of our findings, lines 170-185.

9. Figure 3: n=3 replicates from two experiments for these studies are mentioned in material and methods appear to be low to show significance.

We added more clarification to the methods section by stating the number of repeats for each assay. Each result is representative of no less than four independent experiments. Experiments were repeatable and consistent, with each graph deriving from two or more separate assays and six or more replicates. Further, the robustness of the data is evidenced by new data in Fig. 4 using two different G4 ligands for

each antibody in the CHIP assays, meaning that each antibody (BG4, γ H2AX, or RAD51) was tested in two different G4-ligand treatments, so there are four or more independent experiments for each antibody.

10. Page 4, line 154 and 156 and , IVS1 is written instead of IVS21.

This was indeed a repeat unit from intron 1, it is a poly-purine repeat “GGGAGGG” similar to those found in intron 21. Clarification was added to the methods, and a better description is included on line 62.

REVIEWER COMMENTS

Reviewer #1 (Remarks to the Author):

I would like to thank the authors for carefully adjusting their manuscript according to the referees comments. The manuscript has significantly improved and the story is a lot better shaped and structured. However there are still some remaining questions that needs to be addressed prior publication

1. Still it remains not clear, why they have selected from intron one (where 16 G4 motifs are present) only one G4 for in vitro analysis. I would assume all predicted should form. Here at least three G4 motifs should be addressed. Further if the point is that mouse has little predicted G4s, overall numbers of folded G4s in both human and mouse (they can use published experimental data sets) are required. How conserved is this region from mouse to humans?

2. I really like this co-staining experiments. CASFISH data are very beautiful and highly strengthen their conclusion. However if the conclusion is that there is no overlap in mouse, this work needs to be done also in mouse showing no G4 stainings at the PKD1 locus. Similar to the ChIP experiment

3. Further the newly added ChIP experiments are very good. They also are inline with the conclusion that G4 form at the PKD1 in human but not in mouse. In mouse they see no enrichment of G4s at PKD1, however it needs to be shown that the PhenDC works in these cell lines. Do G4 enrich at other loci?

Based on this the conclusion is that G4 form in hPKD1. As there are also G4 motifs in mPKD1 I would assume also there G4 will form. This needs to be clarified or shown that there are really non in mouse mPKD1.

4. The data linking G4 and DNA damage are very good and supporting the model that DNA breaks a driver for the PKD1 inactivation mechanisms. If this strong conclusion is correct, do they observe mutation signature more in the PKD1 gene at G4 motifs than in neighboring non G4 region of the PKD1 gene? Here they could check published mutational signatures. This experiment will give already first indication to the next questions that I think are important to address. Do all of the predicted hPKD1 form in cells? Which are the relevant once that cause the DNA damage and why? And why are the G4 stabilized in the disease, which helicases are mutated that normally control G4 formation?

Reviewer #2 (Remarks to the Author):

The authors have addressed each of the concerns raised and their responses have greatly improved the manuscript. The CASFISH experiments are particularly commendable. Some minor concerns remain about clarity regarding the details of the experimental approaches and the interpretation of some results. Individual concerns are detailed below. If these can be addressed, the manuscript is suitable for publication.

1. The revised title is excellent, but it would be better as “potential second hit mechanism” since LOH and gene inactivation by G-4 formation have not yet been demonstrated directly.
2. The cropped western blots in supplementary Figure 2 should include day 14 and an explanation regarding the bright band at 50 kDa is needed – is this a non-specific band?
3. Given the reduction in background achieved in dot blots by using tween, a question remains whether tween was used when washing the stained tissues, and if not, why.
4. In the Figure 1 legend, it should be stated explicitly whether the tissue in panels D and E are from APKD patients.
5. How ChIP enrichment values were calculated is still not stated as clearly as it could be. For example, were CT values from an input sample subtracted from the CT values of the respective IP sample to generate deltaCT values for each “beads alone” and antibody IP sample? Was an input sample included in the calculations to normalize the amount of genomic DNA in each sample? If not, what steps were taken to support an assumption that the amount of DNA was equal?

Reviewer #3 (Remarks to the Author):

Reviewer #4 (Remarks to the Author):

In this revised manuscript the authors have addressed my concerns. There are few more queries that I have from the revised version:

-In material and methods, line 270, "Frozen non-fixed ADPKD tissue was a generous gift from the University of Kansas Medical 271 Center provided by the NIDDK sponsored (NIH DK126126) Polycystic Kidney Disease Research 272 Resource Consortium. Five micron-thick kidney sections were rehydrated with PBS for 10 273 minutes and treated with 0.5 µg/ml RNase A for 30 minutes to destroy G4-RNAs." Not clear how the sections were derived from frozen unfixed tissues?

- In Fig 4a and line 107, it is mentioned that hPKD1 mRNA abundance was significantly ($P < 0.05$) reduced by more than half after two days..... 3.6 times after a week---Please indicate all P values in the figure itself at each point for clarity, not clear what the p values was after a week.

The authors thank the reviewers for useful suggestions, all comments have been addressed. Modifications to the manuscript are indicated with **red text**. Specific responses are listed below for each reviewer comment, in *red italics*.

REVIEWER COMMENTS

Reviewer #1 (Remarks to the Author):

I would like to thank the authors for carefully adjusting their manuscript according to the referees comments. The manuscript has significantly improved and the story is a lot better shaped and structured. However there are still some remaining questions that needs to be addressed prior publication

1. Still it remains not clear, why they have selected from intron one (where 16 G4 motifs are present) only one G4 for in vitro analysis. I would assume all predicted should form. Here at least three G4 motifs should be addressed. *The 16 G4 motifs from IVS1 are a tandem repeat, so the same sequence. CD scans for three additional PKD1 G4 motifs, for a total of four, are now included (Suppl. Fig. 1).*

Further if the point is that mouse has little predicted G4s, overall numbers of folded G4s in both human and mouse (they can use published experimental data sets) are required. How conserved is this region from mouse to humans?

This is present in the text, line 61. Results for G4 prediction agree with the trend observed for published datasets for experimentally validated G4s curated by Endoquad for human and mouse PKD1. There do not appear to be G4 DNAs in mPkd1 relevant to DNA breaks and second hit, we further emphasize that point with new text on line 128.

2. I really like this co-staining experiments. CASFISH data are very beautiful and highly strengthen their conclusion. However if the conclusion is that there is no overlap in mouse, this work needs to be done also in mouse showing no G4 stainings at the PKD1 locus. Similar to the ChIP experiment

We agree, CASFISH results here are confined to the human gene and we make no conclusions regarding the mouse ortholog in this experiment. Mouse Pkd1 G4 results are described using a much more quantitative approach (BG4-ChIP).

Data addressing this concern is further shown in Figure 4. Since G4 ligands do not elicit breaks in mPKD1 (Figure 4b, c) or significantly lower expression (Figure 4a), there must be a lack of physiologically relevant G4 DNAs in the mouse gene. We added new text to better emphasize this point on line 128.

3. Further the newly added ChIP experiments are very good. They also are inline

with the conclusion that G4 form at the PKD1 in human but not in mouse. In mouse they see no enrichment of G4s at PKD1, however it needs to be shown that the PhenDC works in these cell lines. Do G4 enrich at other loci?

Both ligands are functional in mIMCD3. New ChIPs show that the G4-rich Sgamma3 locus in mIMCD3 is enriched compared to mPKD1 in the presence of either PhenDC3 or CX-5461, Supplementary Fig. 3.

Based on this the conclusion is that G4 form in hPKD1. As there are also G4 motifs in mPKD1 I would assume also there G4 will form. This needs to be clarified or shown that there are really non in mouse mPKD1.

Results from Figure 4 (DNA break ChIPs) indicate the absence of physiologically relevant G4 DNAs in mPkd1. We added a sentence to further emphasize that result, line 128.

4. The data linking G4 and DNA damage are very good and supporting the model that DNA breaks a driver for the PKD1 inactivation mechanisms. If this strong conclusion is correct, do they observe mutation signature more in the PKD1 gene at G4 motifs than in neighboring non G4 region of the PKD1 gene? Here they could check published mutational signatures. This experiment will give already first indication to the next questions that I think are important to address. Do all of the predicted hPKD1 form in cells? Which are the relevant once that cause the DNA damage and why? And why are the G4 stabilized in the disease, which helicases are mutated that normally control G4 formation?

We are glad the conclusions are considered strong. Connecting somatic mutations to a specific G4 structure or helicase deficiencies is beyond the scope of this study but certainly a logical next step for subsequent research. We however did map somatic mutations (published by the Rennert group) onto hPKD1 G4s, Supplementary Figure 5., and discussed the implications and future directions on lines 170-180.

Reviewer #2 (Remarks to the Author):

The authors have addressed each of the concerns raised and their responses have greatly improved the manuscript. The CASFISH experiments are particularly commendable. Some minor concerns remain about clarity regarding the details of the experimental approaches and the interpretation of some results. Individual concerns are detailed below. If these can be addressed, the manuscript is suitable for publication.

1. The revised title is excellent, but it would be better as “potential second hit

mechanism” since LOH and gene inactivation by G-4 formation have not yet been demonstrated directly.

We are pleased that the revised title is an excellent improvement.

2. The cropped western blots in supplementary Figure 2 should include day 14 and an explanation regarding the bright band at 50 kDa is needed – is this a non-specific band?

Uncropped Western with day 14 is now shown. The bands are expected, polycystin-1 is normally processed into soluble fragments. This is now mentioned in the legend.

3. Given the reduction in background achieved in dot blots by using tween, a question remains whether tween was used when washing the stained tissues, and if not, why.

CASFISH was optimized for signal specificity and cell continuity of the fragile tissue sections. We found that tween disrupted tissue morphology and PBS alone was sufficient for washes of CASFISH of frozen tissue. This is now explained in the methods, line 298-299.

4. In the Figure 1 legend, it should be stated explicitly whether the tissue in panels D and E are from APKD patients.

Clarification is added for supplementary figure 2, panels D/E, which are ADPKD tissue.

5. How ChIP enrichment values were calculated is still not stated as clearly as it could be. For example, were CT values from an input sample subtracted from the CT values of the respective IP sample to generate deltaCT values for each “beads alone” and antibody IP sample? Was an input sample included in the calculations to normalize the amount of genomic DNA in each sample? If not, what steps were taken to support an assumption that the amount of DNA was equal?

Input was equal. We added further detail to better explain in the methods, lines 330-34.

Reviewer #3 (Remarks to the Author):

Reviewer #4 (Remarks to the Author):

In this revised manuscript the authors have addressed my concerns. There are few more queries that I have from the revised version:

-In material and methods, line 270, "Frozen non-fixed ADPKD tissue was a generous gift from the University of Kansas Medical 271 Center provided by the NIDDK sponsored (NIH DK126126) Polycystic Kidney Disease Research 272 Resource Consortium. Five micron-thick kidney sections were rehydrated with PBS for 10 273 minutes and treated with 0.5 µg/ml RNase A for 30 minutes to destroy G4-RNAs." Not clear how the sections were derived from frozen unfixed tissues?

They were OTC embedded, methods were adjusted to include that, line 283.

- In Fig 4a and line 107, it is mentioned that hPKD1 mRNA abundance was significantly ($P < 0.05$) reduced by more than half after two days..... 3.6 times after a week---Please indicate all P values in the figure itself at each point for clarity, not clear what the p values was after a week.

P values were added to Figure 4a.

REVIEWERS' COMMENTS

Reviewer #1 (Remarks to the Author):

We thank the author for carefully addressing all the concerns of the referees. They have pleased my concerns and I have no further comments

Reviewer #2 (Remarks to the Author):

The authors have addressed all comments satisfactorily.

Reviewer #3 (Remarks to the Author):
